# Deciphering Selectivity Mechanism of BRD9 and TAF1(2) toward Inhibitors Based on Multiple Short Molecular Dynamics Simulations and MM-GBSA Calculations

**DOI:** 10.3390/molecules28062583

**Published:** 2023-03-12

**Authors:** Lifei Wang, Yan Wang, Yingxia Yu, Dong Liu, Juan Zhao, Lulu Zhang

**Affiliations:** School of Science, Shandong Jiaotong University, Jinan 250357, China

**Keywords:** BRD9, TAF1(2), MSMD, binding selectivity, free energy landscapes

## Abstract

BRD9 and TAF1(2) have been regarded as significant targets of drug design for clinically treating acute myeloid leukemia, malignancies, and inflammatory diseases. In this study, multiple short molecular dynamics simulations combined with the molecular mechanics generalized Born surface area method were employed to investigate the binding selectivity of three ligands, 67B, 67C, and 69G, to BRD9/TAF1(2) with IC_50_ values of 230/59 nM, 1400/46 nM, and 160/410 nM, respectively. The computed binding free energies from the MM-GBSA method displayed good correlations with that provided by the experimental data. The results indicate that the enthalpic contributions played a critical factor in the selectivity recognition of inhibitors toward BRD9 and TAF1(2), indicating that 67B and 67C could more favorably bind to TAF1(2) than BRD9, while 69G had better selectivity toward BRD9 over TAF1(2). In addition, the residue-based free energy decomposition approach was adopted to calculate the inhibitor–residue interaction spectrum, and the results determined the gatekeeper (Y106 in BRD9 and Y1589 in TAF1(2)) and lipophilic shelf (G43, F44, and F45 in BRD9 and W1526, P1527, and F1528 in TAF1(2)), which could be identified as hotspots for designing efficient selective inhibitors toward BRD9 and TAF1(2). This work is also expected to provide significant theoretical guidance and insightful molecular mechanisms for the rational designs of efficient selective inhibitors targeting BRD9 and TAF1(2).

## 1. Introduction

Acute myeloid leukemia (AML), a morphologically, clinically, and genetically heterogeneous disorder caused by mutations in myeloid differentiation and proliferation, has severely imperiled patients’ lives around the world [1,2,3]. Drug design is the creative process of finding specific small molecules that can obstruct or enhance the functions of biological targets based on the action mechanism of the drug and target [4,5,6,7]. Designing small molecules that inhibit the activity of bromodomain-containing proteins (BRDs) is a promising therapeutic strategy to treat many kinds of diseases, including cancer, inflammation, and cardiovascular and autoimmune diseases [8,9,10,11]. Bromodomain-containing protein 9 (BRD9) belonging to the BRD family, a major constituent of the SWI/SNF chromatin remodeling complex named non-canonical BAF, has been identified as a novel therapeutic target in AML [12,13]. The sensitivity of AML cells is correlated with the level of BRD9, and AML cells endure terminal differentiation and cycle arrest with the degradation of BRD9 [14]. Meanwhile, the second bromodomain of the human transcription initiation factor TFIID subunit 1 (TAF1(2)) is overexpressed in a variety of cancers and plays a significant role in AML1-ETO fusion gene expression [15]. Furthermore, multiple reports indicate the key roles of TAF1(2) in AML and provide a new theoretical structural framework to develop direct-acting small molecule inhibitors of TAF1(2) as prospective inflammation pathophysiology and cancer therapeutics [16,17,18,19,20].

The BRD9 and TAF1(2), as shown in Figure 1, structurally share four left-handed α-helices (αA, αB, αC, αZ) constituting up-and-down four-helix bundles, which form two loops between helices αA and αZ (ZA loop) and αB and αC (BC loop), respectively [21]. These two loops feature a hydrophobic pocket, which frequently generates two stable hydrogen bonds, one is between the amide and asparagine at the top of the BC loop in BRD9/TAF1(2), and the other is water-mediated and formed with the tyrosine situated at the ZA loop of BRD9/TAF1(2) [22]. In addition, the hydrophobic binding pocket of BRD9/TAF1(2) consists of the gatekeeper at the head of the αC helix and the lipophilic shelf with the first three amino acid residues of the ZA loop [23]. In BRD9, three residues, G43, F44, and F45, adjacent to the ZA channel constitute the lipophilic shelf, and Y106 at the beginning of the αC helix is the gatekeeper. Meanwhile, the lipophilic shelf of TAF1(2) is composed of three residues, W1526, P1527, and F1528, and residue Y1589 is regarded as the gatekeeper. Although the tyrosine and asparagine residues are heavily conserved in the vast majority of bromodomain proteins, there are apparent conformational changes in the gatekeeper, lipophilic shelf, and ZA channel residues. Based on the important target roles of BRD9 and TAF1(2) in drug design toward human cancers, Crawford and coworkers solved the crystal structures of 67B- and 69G-bound BRD9, as well as 67B- and 67C-associated TAF1(2) [22]. Despite the highly similar structures shared by 67B, 67C, and 69G (Figure 2C–E), three inhibitors have different binding affinities to BRD9 and TAF1(2), with IC_50_ values of 230/59 nM, 1400/46 nM, and 160/410 nM for BRD9/TAF1(2), respectively. Therefore, it is essential to clarify the binding selectivity of inhibitors to BRD9 and TAF1(2) and the conformational changes of the two proteins caused by inhibitor binding for designing drugs for anti-cancer treatment.

Until now, various simulation approaches, including conventional molecular dynamics (cMD) [24,25,26,27,28,29], multiple short molecular dynamics (MSMD) [30], accelerated molecular dynamics (aMD) [31,32,33], Gaussian accelerated molecular dynamics (GaMD) [34,35,36,37,38], and multiple replica Gaussian accelerated molecular dynamics (MR-GaMD) simulations [39,40,41] have been used to investigate the conformational alterations and binding mechanisms of receptors due to ligand associations and residue mutations [42,43,44]. Several intensive molecular dynamics studies have successfully deciphered the molecular mechanism regarding the binding selectivity of inhibitors toward homological proteins with highly similar tertiary structures [45,46,47,48,49,50,51,52]. In fact, numerous molecular dynamics simulation works were also conducted to research the binding modes of different ligands to BRD9 and TAF1(2). For instance, Wang et al. combined molecular dynamics and binding free energy predictions to determine the binding mechanism of three small molecule ligands, 5SW, 5U2, and 5U6, toward BRD9 and identify the hot interaction spots of the protein with inhibitors [53]. Liu’s group applied MSMD simulations and binding affinity calculations to the investigation of the binding difference of inhibitors to different BRD families, and their results provided useful information for clarifying the selective mechanisms of BRD families [54,55,56]. Song et al. estimated the binding free energies and energetic contributions of individual residues of four pyridinone-like scaffold inhibitors complexed with BRD9 based on cMD simulations, and their results suggested that the aromatic ring and dimethoxyphenyl structure combined into a pyridinone scaffold effectively enhanced the BRD9 binding affinity [57]. Magno et al. performed twenty-four independent MD simulations and free energy profile analyses to investigate the spontaneous and reversible binding of acetylated lysine to TAF1(2), and the obtained dynamical information indicated that hydrogen bond interactions stabilized the two main binding modes of TAF1(2) [58]. Thus, it is important to decipher the binding mode and selectivity mechanism of inhibitors at the atomic levels for the development of potent small-molecule inhibitors targeting BRD9 and TAF1(2).

In the current study, in order to illustrate the selective mechanism of BRD9 and TAF1(2), three small-molecule inhibitors, 67B, 67C, and 69G, were chosen to determine their binding selectivity for BRD9 and TAF1(2) by performing MSMD simulations and binding free energy computations. Encouragingly, MSMD simulations can extract more rational conformational sampling than cMD simulations, which has been verified in previous works [59,60,61]. Furthermore, principal component analysis (PCA) [62,63], dynamics cross-correlation maps (DCCMs), and free energy landscapes (FELs) were combined to investigate the conformational variations and internal dynamics of BRD9 and TAF1(2) caused by inhibitor associations. To contrast the structural differences between BRD9 and TAF1(2), the PyMOL software was utilized to align the two complexes, and their structures are displayed in Figure 2A. As shown in Figure 2B, the binding pockets of BRD9 and TAF1(2) were drawn in surface forms, while the ligand was depicted in stick form. The structures of three small-molecule inhibitors. 67B, 67C, and 69G, are shown in line forms in Figure 2C–E, respectively. The three inhibitors 67B, 67C, and 69G share similar structures, except for the red rectangle. It was conducive to determine the impact of the structural variations in 67B, 67C, and 69G on the binding selectivity of BRD9 and TAF1(2) for the design of small-molecule inhibitors associated with bromodomain proteins. In this work, MSMD simulations and several analysis methods, such as PCA, inhibitor–residue interactions, calculations of DCCMs, analysis of FELs, hydrogen bonding interactions (HBIs), and hydrophobic interactions (HIs), were integrated to carry out this study. In the meantime, this study is also expected to provide rational information for the design of small-molecule inhibitors toward BRD9 and TAF1(2).

## 2. Results and Discussion

### 2.1. Structural Flexibilities and Fluctuations of BRD9 and TAF1(2)

In order to execute reliable and rational conformational samplings of BRD9 and TAF1(2), a total of 1.2 μs MSMD simulations, including three individual cMD simulations of 400 ns, were conducted on the *apo* BRD9 and TAF1(2), as well as the six inhibitor–BRD9 and inhibitor–TAF1(2) systems with three inhibitors, 67B, 67C, and 69G. To assess the overall stability of the MSMD simulations, the fluctuations in the root-mean-square-deviations (RMSDs) of backbone atoms in BRD9 and TAF1(2) relative to the corresponding initial conformations of the six complexes over time were calculated and are depicted in Appendix A. On the whole, the structural fluctuation range of the *apo* BRD9 was higher than that of the inhibitor–BED9 complexes while the structural fluctuation extent of the *apo* TAF1(2) was similar to that of the inhibitor–TAF1(2) complexes (Appendix A). The structural variations of three replicas of 67B-, 67C-, and 69G-associated BRD9/TAF1(2) were convergent after 100 ns of cMD simulations (Appendix A). Hence, the stable portions (100–400 ns) from three independent cMD simulations were concatenated together to create a single integrated trajectory (SIT) of 900 ns for each complex, which was used to execute all computations and dynamics analyses.

To further determine the structural flexibilities of the BRD9 and TAF1(2) induced by inhibitor associations, the root-mean-square fluctuations (RMSFs) of the C_α_ atoms in BRD9 and TAF1(2) were calculated based on the SIT (Figure 3). According to the comparison, BRD9 and TAF1(2) produced similar RMSFs fluctuations, suggesting that BRD9 and TAF1(2) embodied common rigid and flexible domains. For the BRD9-related systems, the binding of three inhibitors weakened the structural flexibility of BRD9, especially for the ZA-loop (Figure 3A,B). However, for the TAF1(2)-related systems, the binding of 67B and 67C slightly reduced the structural flexibility of TAF1(2), while the presence of 69G strengthened that of TAF1(2), particularly for the ZA-loop (Figure 3C,D). Obvious structural alterations mainly existed in four regions, including L1 (residues 36–45 for BRD9 and 1519–1528 for TAF1(2)), L2 (residues 49–67 for BRD9 and 1532–1550 for TAF1(2)), L3 (residues 74–85 for BRD9 and 1557–1568 for TAF1(2)), and L4 (residues 99–110 for BRD9 and 1582–1593 for TAF1(2)). The lipophilic shelf contained three residues (G43, F44, and F45 in BRD9, and W1526, P1527, and F1528 in TAF1(2)) at the end of the region L1, while the gatekeeper included one residue (Y106 in BRD9, and Y1589 in TAF1(2)) at region L4. For BRD9, the inhibitor associations induced evident alterations in the structural flexibility at regions L1 and L2 (Figure 3A); however, the bindings of inhibitors with TAF1(2) only yielded an obvious influence on region L2 (Figure 3C). These alterations in RMSFs indicated that the structural flexibility of BRD9 was higher than that of TAF1(2). The RMSF values of the 67B–, 67C–, and 69G–BRD9 compounds in region L3 and the corresponding ones in region L4 of TAF1(2) were lower than 1.0 Å, demonstrating that the two regions were rigid. However, the RMSFs values of region L2 in BRD9 and TAF1(2) were above 1.5 Å, suggesting that region L2 was flexible. Owing to 67B and 67C binding, the main portions of RMSFs in BRD9 were obviously larger than the corresponding ones in TAF1(2). In the bound state of 69G, the flexibility of three regions, L1, L2, and L3, in BRD9 was lower than those of TAF1(2), while the flexibility of region L4 in BRD9 was slightly higher than that of TAF1(2). The results signify that several residues of the four aforementioned regions were potential central factors driving the selective binding of inhibitors toward these two proteins.

As shown in Figure 3B, G43 was the first residue of ZA channel, Y106 was the gatekeeper of BRD9, W1526 was the first residue of ZA channel, and Y1589 was the corresponding gatekeeper in TAF1(2). The distances between the first residue of the ZA channel and the gatekeeper (Y106–G43 in BRD9 or Y1589–W1526 in TAF1(2)) were determined from the SIT, and the corresponding frequency distributions are displayed in Figure 4A,C,E. Moreover, the frequency distributions of the Chi dihedral angle of the side chain (Y106 in BRD9 and Y1589 in TAF1(2)) are depicted in Figure 4B,D,F. As shown in Figure 4A, the higher peak values of the Y106 C_α_–G43 C_α_ distance of the BRD9–67B complex were distributed range between 11.1 and 12.0 Å, while the higher peak of the Y1589 Cα–W1526 Cα distance of the TAF1(2)–67B complex was at 10.2 Å. The above results demonstrate that the distance between the gatekeeper and the ZA channel in BRD9 was slightly larger than that in TAF1(2), which suggested that the hot spot site volume of TAF1(2) was smaller than that of BRD9. The peak values of the Chi dihedral angle in the side chain of Y106 for the 67B–, 67C–, and 69G–BRD9 complexes were 313.3°, 310.0°, and 308.0°, respectively, while the ones for the 67B–, 67C–, and 69G–TAF1(2) complexes were 297.8°, 299.7°, and 301.8°, separately. The results show that the hot spots of BRD9 had higher flexibility than those of TAF1(2), which is consistent with the above fluctuations in the RMSFs. 

### 2.2. Internal Dynamics of BRD9 and TAF1(2)

To explore the changes in the internal dynamics of BRD9 and TAF1(2) due to inhibitor associations, the cross-correlation coefficients were calculated by using the *C_α_* atomic coordinates recorded in the SIT, and cross-correlation maps are displayed in Figure 5. According to the color-decoded patterns, the negative regions (dark blue and plain blue) indicated extremely anticorrelated (AC) movements, while the highly positive regions (red and yellow) were related to strongly positively correlated (PC) motions between specific residues. As shown in Figure 5, the binding of 67B, 67C, and 69G produced evident influences on the motion modes of BRD9 and TAF1(2).

For BRD9 associated with 67B, 67C, and 69G (Figure 5A,C,E), regions R1, R2, and R3 generated significant AC motions. Compared with BRD9 complexed with 67B, 67C, and 69G, the binding of 67B, 67C, and 69G to TAF1(2) reduced the AC motions in regions R1, R2, and R3 (Figure 5B,D,F). The results demonstrate that the bindings of identical inhibitors yielded distinct influences on the internal dynamics of BRD9 and TAF1(2), which signified that important residues located in R1–R3 of BRD9 and TAF1(2) may have yielded significant hydrophobic and hydrogen bonding interactions with inhibitors and played critical roles in the binding selectivity of ligands toward BRD9 and TAF1(2).

In addition, principal component analysis (PCA) was conducted to decode the conformational alterations of BRD9 and TAF1(2) due to the associations with 67B, 67C, and 69G, respectively, and the function of forty eigenvalues stemming from the diagonalization of covariance matrix versus the related eigenvector indexes is displayed in Figure 6. The first few larger eigenvalues represent the primarily collective motions of the structural domain in these two proteins. The first six eigenvalues accounted for 79.07%, 72.72%, and 60.36% of the total movements for the 67B–, 67C–, and 69G–BRD9 complexes and 70.37%, 75.08%, and 75.07% of the total motions of the 67B–, 67C–, and 69G–TAF1(2) complexes, respectively.

To gain more insight into the alterations in motional modes between BRD9 and TAF1(2) induced by inhibitor associations, the first eigenvectors of six systems were visualized in six porcupine plots (Appendix A). The direction of the arrow reflects the direction of the movements and the length of the arrow represents the strength of the motions. In contrast with 67B-, 67C-, and 69G-bound BRD9, the bindings of these inhibitors not only altered the movement directions of the L2, L3, and L4 in TAF1(2), but also altered the movement amplitude of these three loops. Furthermore, the αZ helix of the 67B–BRD9 complex moved toward the left and down (Appendix A), while that of the 67B–TAF1(2) was altered toward the right and up (Appendix A). The αZ helix of the 67C–BRD9 complex moved toward the right (Appendix A), while that of the 67C–TAF1(2) was transformed toward the left and down (Appendix A). The αZ helix of the 69G–BRD9 complex moved upward (Appendix A), but that of the 69G–TAF1(2) was altered toward the left and down (Appendix A). The above discussions indicate that the conformational alterations of BRD9 and TAF1(2) extracted from MSMD simulations probably led to the distinct binding selectivities of 67B, 67C, and 69G toward BRD9 and TAF1(2).

### 2.3. Binding Ability of Inhibitors to BRD9 and TAF1(2)

To further evaluate the variance in the binding abilities of 67B, 67C, and 69G to BRD9 and TAF1(2), the MM-GBSA approach was employed to calculate the binding free energies (BFEs) of three ligands to BRD9 and TAF1(2) by using 300 conformational structures withdrawn from the 900 ns SIT with a time step of 3 ns. Fifty structural frames were taken from the above 300 conformations at an interval of 6 conformations to calculate the contributions of entropy (−TΔS) to the binding associations through the normal mode analysis approach. All energetic components resulting from the MM-GBSA calculations are listed in Table 1. The ranks of the BFEs of 67B–BRD9/TAF1(2), 67C–BRD9/TAF1(2), and 69G–BRD9/TAF1(2) were consistent with those of the experimental values, which demonstrated that the computed BFEs were reliable and rational. The energies with positive values provided unfavorable factors for inhibitor associations, while the negative components contributed favorable forces for inhibitor bindings.

As shown in Table 1, for BRD9 and TAF1(2) bound by 67B, 67C, and 69G, the negative electrostatic interaction energies ($\Delta E_{ele}$) were absolutely overwhelmed by positive polar solvation energies ($\Delta G_{gb}$) to form unfavorable terms ($\Delta G_{ele+gb}$) for inhibitor associations. The unfavorable factors of the entropy changes (−TΔS) also weakened the binding associations of 67B, 67C, and 69G to BRD9 and TAF1(2). Meanwhile, the negative values of the nonpolar solvation energies ($\Delta G_{nonpol}$) and Van der Waals interactions ($\Delta E_{vdW}$) produced favorable factors ($\Delta E_{vdW+nonpol}$) for the inhibitor–BRD9/TAF1(2) bindings. As shown in Table 1, the value of $\Delta G_{ele+gb}$ for 67B–TAF1(2) was reduced by 0.90 kcal/mol relative to that of 67B–TAF1(2), and the favorable term$\;\Delta E_{vdW+nonpol}$ of 67B–TAF1(2) was enhanced by 2.77 kcal/mol in comparison with that of 67B–BRD9, which led to an increase of 3.67 kcal/mol in the enthalpy changes of the 67B–TAF1(2) compared with that of 67B–BRD9. Moreover, the value of −TΔS for 67B–TAF1(2) was strengthened by 0.94 kcal/mol relative to that of 67B–BRD9. Overall, the binding affinity of 67B–TAF1(2) was enhanced by 2.73 kcal/mol, suggesting that 67B generated a stronger association with TAF1(2) than with BRD9. In the case of 67C, an increase of 3.41 kcal/mol in the $\Delta E_{vdW+nonpol}$ of the 67C–TAF1(2) complex compared with that of 67C–BRD9 led to an increase in the enthalpy changes of the 67C–TAF1(2) complex compared with that of 67C–BRD9. The entropy changes −TΔS of the 67C–TAF1(2) complex decreased by 1.94 kcal/mol relative to that of 67C–BRD9, which finally resulted in an increase of 4.99 kcal/mol in the binding free energy of the 67C–TAF1(2) complex relative to that of 67C–BRD9. Therefore, the binding affinity of 67C to TAF1(2) was higher than that for BRD9. For ligand 69G, the value of $\Delta G_{ele+gb}$ of the 69G–TAF1(2) complex was decreased by 0.6 kcal/mol compared with that of the 69G–BRD9 complex, and the negative component $\Delta E_{vdW+nonpol}$ of the 69G–TAF1(2) complex was decreased by 2.94 kcal/mol relative that of 69G–BRD9, which resulted in an overall decrease of 2.34 kcal/mol in the binding enthalpy of the 69G–TAF1(2) complex relative to that of 69G–BRD9. Furthermore, the −TΔS of the 69G–TAF1(2) complex was decreased by 0.57 kcal/mol compared with that of 69G–BRD9. In view of the above two factors, the binding ability of 69G to BRD9 was increased by 1.77 kcal/mol relative to TAF1(2), suggesting that 69G produces more favorable binding to BRD9 than TAF1(2).

### 2.4. Binding Selectivity Probed by Ligand–Residue Interactions

To further explicate the binding selectivity of three inhibitors, 67B, 67C, and 69G, to BRD9 and TAF1(2), the ligand–residue interaction spectrum was estimated with the residue-based free-energy decomposition approach. Appendix A provides the energetic contributions from the backbone and sidechain of critical residues in BRD9 and TAF1(2) associated with 67B, 67C, and 69G. The results indicate that energetic contributions from the sidechain of residues played significant roles in inhibitor–residue interactions. The critical residues of BRD9 and TAF1(2) that constituted vital inhibitor–residue interactions with energetic contributions stronger than 1.0 kcal/mol are displayed in Figure 7, Figure 8, Figure 9, Appendix A. Moreover, the CPPTRAJ software was used to recognize the hydrogen bonding interactions (HBIs) of 67B, 67C, and 69G with BRD9 and TAF1(2) (Table 2). The structural information of the hydrogen bonds and the relevant radial distribution functions (RDF) of the H–O distance between the three inhibitors and key residues of BRD9 and TAF1(2) are shown in Figure 10 and Figure 11.

#### 2.4.1. Bound BRD9 against the 67B-Bound TAF1(2) 

The interaction energies of 67B with F44, V49, I53, N100, and Y106 in BRD9 were −2.04, −1.71, −1.87, −2.82, and −2.22 kcal/mol, respectively, and they structurally arose from the CH–π interactions of the alkyls in three residues, V49, I53, and N100, with the hydrophobic ring of 67B and the π−π interactions of the hydrophobic rings in two residues, F44 and Y106, with the corresponding rings in 67B (Figure 7A, Figure 8A, Figure 9A and Appendix A). Meanwhile, 67B formed two HBIs with BRD9, containing 67B-O1···Asn100-ND2-HD21 and Asn100-OD1···67B-N2-H12 with occupancies of 94.64% and 91.54% in Table 2, respectively. In contrast with the 67B–BRD9 complex, the hydrophobic interactions and HBIs of 67B with TAF1(2) were highly similar to those of the 67B–BRD9. As shown in Figure 8A, the interaction energy alterations of 67B with residues (G43 and W1526), (V49 and V1532), (T50 and N1533), (I53 and F1536), (A54 and V1537), (A96 and S1579), and (Y106 and Y1589) in (BRD9 and TAF1(2)) were above 0.6 kcal/mol, suggesting that the seven residues played significant roles in the binding selectivity of 67B to BRD9 and TAF1(2).

#### 2.4.2. 67C-Bound BRD9 Versus the 67C-Bound TAF1(2)

The 67C yielded binding energies stronger than −1.0 kcal/mol with five residues, F44, V49, I53, A96, N100, and Y106, of BRD9 (Figure 7C). The interaction strength of 67C with F44 and Y106 was scaled by −2.13, and −1.49 kcal/mol, which structurally agreed with the π−π interactions between the rings of F44 and Y106 with those of ligand 67C. The binding energy values of residues V49, I53, A96, and N100 with 67C were −2.30, −1.98, −1.0, and −3.06 kcal/mol, respectively, and they were mostly provided by the CH–π interactions between the CH groups of these four residues and the ring of 67C (Figure 9C). The frequency distribution of the distances between 67C and the critical residues of BRD9 is displayed in Figure 9D, and shows that the above-mentioned CH–π and π–π interactions were highly stable. As shown in Table 2 and Figure 10C,D, Thr50 and Asn100 in BRD9 formed three HBIs with 67C, namely 67C–O1····Asn100-ND2-HD21, 67C-O····Thr50-N-H, and Asn100-OD1····67C-N2-H7, with occupancies of 98.72%, 29.81%, and 94.89%, respectively. Compared with the 67C–BRD9 complex, the variation in the interactions of 67C with residues (G43 and W1526), (F44 and P1527), (F45 and F1528), (I53 and F1536), (A54 and V1537), and (A96 and S1579) in (BRD9 and TAF1(2)) was above 0.49 kcal/mol, suggesting that these residues played key roles in the binding selectivity of 67C to BRD9 and TAF1(2).

#### 2.4.3. The 69G-Bound BRD9 over the 69G-Bound TAF1(2) 

As shown in Figure 7E, the favorable binding energies of 69G with six residues in BRD9, i.e., F44, V49, I53, A96, N100, and Y106, were above −1.0 kcal/mol. For the binding of 69G to BRD9, F44 and Y106 contributed interactions of −2.79 and −2.28 kcal/mol, which structurally agreed with the π−π hydrophobic interactions between the ring of 69G and those of F44 and Y106 (Figure 9E,F). As shown in the geometric conformations (Figure 9E), the CH groups of V49, I53, A96, and N100 of BRD9 were adjacent to the hydrophobic ring of 69G. Figure 7E indicates that V49, I53, A96, and N100 generated binding energies of −1.8, −1.86, −1.11, and −2.86 kcal/mol, respectively, for the 69G–BRD9 complex. Furthermore, 69G formed two robust HBIs with BRD9, containing 69G-O15···Asn100-ND2-HD21, and Asn100-OD1···69G-N11-H12, and their occupancies were 99.94% and 78.93%, respectively (Table 2 and Figure 11E,F). According to Figure 7F, Appendix A, Figure 11E,F and Table 2, the binding interactions of 69G with TAF1(2) involved seven favorable interactions with energies stronger than 1.0 kcal/mol and two HBIs. The binding energy variance of 69G with residues (F44 and P1527), (F45 and F1528), (V49 and V1532), (I53 and F1536), (A54 and V1537), and (A96 and S1579) in (BRD9 and TAF1(2)) was higher than 0.41 kcal/mol, indicating that these six residues played significant roles in the binding selectivity of 69G to BRD9 and TAF1(2).

### 2.5. Alterations in the Free Energy Landscapes of BRD9 and TAF1(2) Caused by Inhibitor Bindings

The Free energy landscapes (FELs) can effectively represent various free energy states relative to the conformational alterations of proteins due to changes in the binding environment [64,65,66]. To study the influences of small molecular inhibitors’ associations on the conformational alterations of BRD9 and TAF1(2), projections of the SIT onto the first two principal components arising from the diagonalization of the covariance matrix were used as reaction coordinates to construct the FELs of BRD9 and TAF1(2). The results and structures are displayed in Figure 12, Figure 13 and Appendix A, in which the symbols V1, V2, V3, and V4, and the red points denote different energy valleys recognized by MSMD simulations. As shown in these figures, the associations of three inhibitors, 67B, 67C, and 69G, with BRD9 and TAF1(2) evidently affected the FELs and generated conformational alterations.

#### 2.5.1. The 67B-Associated BRD9 against the 67B-Bound TAF1(2)

The MSMD simulations captured two distinct energy valleys, V1 and V2, and according to the color bar, the typical structure V2 was located at a deeper potential basin than structure V1 (Figure 12A), indicating that the conformations of the 67B–BRD9 complex were primarily distributed at two energetic spaces. The structures of 67B in two typical conformations V1 and V2 of the 67B-associated BRD9 are aligned together in Figure 12B. The superimpositions of two representative structures corresponding to energy valleys V1 and V2 reveal that three domains, L1, L2, and L4, significantly deviated from each other (Figure 12B), suggesting that 67B generates an evident effect on the structural flexibility of the 67B–BRD9 complex. The results showed that 67B had two various binding poses in BRD9, which significantly influenced the association of 67B to BRD9. Regarding the 67B-associated TAF1(2), two distinct energy valleys, V1 and V2, were identified by using entire MSMD simulations, and the color bar indicated that the depth of energy valley V1 was deeper than that of energy valley V2 in Figure 13A, indicating that the structures of the 67B–TAF1(2) complex were mainly distributed at two energetic spaces. The alignments of two conformations located at energy valleys V1 and V2 demonstrated that domains L1 and L2 produced evident deviations from each other (Figure 13B), which possibly generated significant effects on the association of 67B with TAF1(2). The conformational superimpositions of two representative structures of the 67B–TAF1(2) complex in energy valleys V1 and V2 denoted that the association poses of 67B in TAF1(2) yielded primarily parallel slides from each other, which exerted a specific influence on the binding of 67B to TAF1(2).

#### 2.5.2. The 67C-Associated BRD9 versus the 67C-Bound TAF1(2) 

The MSMD simulations detected four primary energy valleys, V1, V2, V3, and V4, in the 67C-associated BRD9; in accordance with the color bar, the depth of energy valley V1 was the deepest in these four energy valleys (Figure 12C), suggesting that the structures of the 67C–BRD9 complex were chiefly populated at four energetic spaces. Four typical structures located at valleys V1, V2, V3, and V4 are superposed in Figure 12D, and the results demonstrate that loops L1 and L2 significantly diverged from each other, indicating that these two loops displayed a considerable structural flexibility and played an essential role in the association of 67C with BRD9. As shown in the conformational alignment of 67C in the typical structures V1, V2, V3, and V4 (Figure 12D), 67C had four distinct association poses and produced large deviations. For the 67C–TAF1(2) complex, three energy valleys, V1, V2, and V3, were recognized by the entire MSMD simulations (Figure 13C), demonstrating that the 67C–TAF1(2) complex was distributed at three conformational subspaces. The alignments of three typical conformations situated at energy basins V1, V2, and V3 denote that domains L1 and L2 yielded evident distortions from each other (Figure 13D). Active regions L1 and L2 yielded obvious distortions among three typical conformations, which probably brought significant effects on the poses of 67C. This result is supported by the apparent structural slides and torsions of 67C shown in Figure 13D. Therefore, the conformational alterations of L1 and L2, and slides and torsions of 67C certainly exerted evident influences on the binding selectivity of 67C to BRD9 and TAF1(2).

#### 2.5.3. The 69G-Bound BRD9 over the 69G-Associated TAF1(2) 

Two energy valleys, V1 and V2, were captured by the MSMD simulations of the 69G–BRD9 complex and on the basis of the color bar; structure V2 was located at a deeper valley than structure V1 (Figure 12E), indicating that the 69G–BRD9 complex occupied two primary energetic spaces. The superimpositions of two representative conformations located at energy valleys V1 and V2 suggested that loop L1 had higher structural flexibility and apparently diverged from each other in the 69G–BRD9 system (Figure 12F). The conformations of 69G in two representative basins, V1 and V2, were superimposed (Figure 12F) and the results indicate that 69G yielded apparent sliding, which indicates a significant influence on the associations of 69G with BRD9. For the 69G-bound TAF1(2), four energy valleys, V1, V2, V3, and V4, were identified by the whole MSMD simulation, and in accordance with the color bar, the depth of the energy valleys in state V3 was deeper than that of the other three energy valleys, V1, V2, and V3 (Figure 13E), indicating that the conformations of the 69G–TAF1(2) were primarily clustered into four energetic spaces. The alignments of the 69G–TAF1(2) complex located at energy valleys V1, V2, V3, and V4 indicated that loops L1 and L2 produced obvious deviations from each other (Figure 13F). The conformations of 69G in four typical structures of the 69G–TFA1(2) complex were superimposed (Figure 13F), and the results demonstrated that 69G contained four distinct association poses and produced evident distortions, which obviously affected the association of 69G with TAF1(2). Therefore, the changes in the orientations and conformations explicitly influenced the binding selectivity of ligand 69G toward BRD9 and TAF1(2).

## 3. Materials and Methods

### 3.1. Modeling Simulated Systems

The initial configurations of the crystal structures were obtained from the Protein Data Bank (PDB): 5I7X and 5I7Y corresponded to the 67B– and 69G–BRD9 complexes, respectively, while 5I29 and 5I1Q corresponded to the 67B– and 67C–TAF1(2) complexes, respectively [22]. Meanwhile, as the crystal structures of the 67C–BRD9 and 69G–TAF1(2) complexes were unavailable, the crystal structures 517Y were superimposed with 5I1Q to generate the configuration of the 670C–BRD9 compound by deleting 69G and TAF1(2) through the PyMol software (https://www.pymol.org accessed on 5 February 2023). Similarly, the crystal structures of the 69G–BRD9 (5I7Y) and 67C–TAF1(2) (5I1Q) complexes were aligned together to produce the initial atomic coordinates of the 69G–TAF1(2) compound by eliminating BRD9 and 67C. Owing to the difference in the length of the residues from BRD9 and TAF1(2), residues 22–102 in BRD9 and residues 1504–1604 in TAF1(2) were used as the starting crystal structures of the MSMD simulations. All missing hydrogen atoms in the crystal structures were added to their corresponding heavy atoms with the Leap module of Amber 20.0 [67,68]. The ff19SB force field [69] and TIP3P model [70] were used to generate the force field parameters of proteins BRD9 and TAF1(2), water molecules, and counter ions. The configurations of three inhibitors, 67B, 67C, and 69G, were optimized using the semi-empirical AM1 approach, and then the atomic BCC charges of 67B, 67C, and 69G were produced by using the Antechamber module in Amber 20.0. The general AMBER force field (GAFF) was applied to generate the force field parameters of three inhibitors, 67B, 67C, and 69G [71]. Four chlorine ions (Cl^−^) and eleven sodium ions (Na^+^) were deposited around the ligand-associated BRD9 and TAF1(2) to generate six neutral simulated systems in the salt environment of 0.15 M NaCl. In addition, octahedral periodic water boxes with 12.0 Å using the TIP3P model were utilized to solvate the ligand–BRD9 or TAF1(2) complexes, and the number of water molecules was about 7600. 

### 3.2. Multiple Short Molecular Dynamics (MSMD) Simulations

pmemd.cuda embedded in Amber 20 was used to implement all of the multiple short-molecular dynamics simulations [72]. To relax each system, a 2500-step steepest descent minimization, another 2500-step conjugate gradient minimization, a 2 ns soft heating process of 0 to 300 K under constant number, volume, and temperature (NVT) condition, and then a 2 ns equilibrium process of 310 K under the constant number, pressure, and temperature (NPT) condition were further carried out. Ultimately, three independent 400 ns cMD simulations were executed with periodic boundary conditions and the particle mesh Ewald (PME) method at a constant temperature (300K) and pressure (1 bar) to relax each simulated complex [73,74]. During the whole MSDM simulations in the present work, the chemical bonds between hydrogen and heavy atoms were constrained with the SHAKE numerical integration algorithm [75]. The temperatures of the simulated complexes were controlled by utilizing the Langevin equation with a collision frequency of 2.0 ps^−1^ with a mollified impulse approach for the Newtonian molecular dynamics [76]. A cutoff value of 10 Å was applied to conduct the estimations of the electrostatic interactions with the PME approach and computations of the van der Waals interactions. In the present work, 1.2 μs MSMD simulations, including three individual cMD simulations of 400 ns, were conducted for the ligand–BRD9/TAF1(2) complexes. To execute the post-process investigation, the equilibrium parts from three independent MSMD trajectories were linked into an SIT. The PCA and DCCMs were executed based on the SIT with the CPPTRAJ module in Amber [77], and the details were introduced in our previous works [78,79].

### 3.3. MM-GBSA Free Energy Computations and Decomposition

Among the methods of binding affinity prediction, although the thermodynamics integration [80,81,82] and free energy perturbation [83,84,85,86,87,88,89] methods can provide more accurate results, these two methods are very expensive in terms of computation time. It is highly important to achieve a good trade-off between accuracy and efficiency in calculations of inhibitor–target binding free energies for drug development, which has been discussed by Rizzuti et al. in their work [90]. Recently, empirical equation-based MM-GBSA and molecular mechanics Poisson Boltzman surface area (MM-PBSA) methods have extensively been used to estimate the binding free energy for numerous ligand–protein and protein–protein interactions [91,92,93,94,95,96,97]. Furthermore, Hou’s group assessed the performance of the MM-GBSA and MM-PBSA approaches by calculating the binding free energies of various biological systems, and their results implied that the MM-GBSA approach could provide more reasonable conclusions [98,99]. Therefore, the MM-GBSA approach was employed to compute the binding free energies of 67B, 67C, and 69G to BRD9 and TAF1(2) with the following Equation (1):(1)$$\Delta G_{bind}=\Delta E_{ele}+\Delta E_{vdW}+\Delta G_{gb}+\Delta G_{nonpol}-T\Delta S$$
where the first two components $\Delta E_{ele}$ and $\Delta E_{vdW}$ denote the electrostatic and van der Waals interactions of 67B, 67C, and 69G with BRD9 and TAF1(2), respectively, which were estimated by using the FF19SB force field. The terms $\Delta G_{gb}$ and $\Delta G_{nonpol}$ indicated the polar and nonpolar solvation free energies, respectively. The third element was solved based on the GB-OBCI model developed by Onufriev et al., and the fourth term was computed using the following empirical formula [100]:(2)$$\Delta G_{nonpol}=\gamma \times \Delta SASA+\beta$$
in which parameters γ and ΔSASA denote the surface tension and the changes in the solvent-accessible surface areas due to inhibitor associations, respectively. Parameters γ and β were to as 0.0072 $\mathrm{kcal}\cdot \mathrm{mol}\cdot {Å}^{-2}$ and 0 $\mathrm{kcal}\cdot {\mathrm{mol}}^{-1}$ in this work, respectively [101].

## 4. Conclusions

Clarifying the binding selectivity of inhibitors to BRD9 and TAF1(2) plays an important role in drug targets for AML, human malignancies, and inflammatory disease therapy. This study aimed to decipher the molecular mechanism of the binding selectivity of three ligands, 67B, 67C, and 69G, toward BRD9 and TAF1(2). MSMD simulations of 1.2 μs, including three independent cMD simulations of 400 ns, were performed on six inhibitor-associated BRD9 and TAF1(2) complexes to decode the binding selectivity of small molecular inhibitors to BRD9 and TAF1(2). The internal dynamics of BRD9 and TAF1(2) were investigated by means of DCCMs, PCA, and FELs, and the results demonstrated that the associations of 67B, 67C, and 69G exerted significant effects on the motion modes of BRD9 and TAF1(2). The BFEs of 67B, 67C, and 69G to BRD9 and TAF1(2) predicated by the MM-GBSA approach indicated that alterations in the binding enthalpy due to the inhibitors’ associations with BRD9 compared with those with TAF1(2) were are primarily responsible for the binding selectivity of ligands to BRD9 over TAF1(2). The results revealed that the enthalpy changes played critical roles in the selectivity recognition of ligands toward BRD9 and TAF1(2), which indicated that 67B and 67C could more favorably bind to TAF1(2) than BRD9, while 69G had better selectivity toward BRD9 versus TAF1(2). The results obtained from the energy contributions of individual residues indicated that three common residues, namely (I53 and F1536), (A54 and V1537), and (A96 and S1579) in (BRD9 and TAF1(2)), generated significant differences in the associations of 67B, 67C, and 69G with BRD9 and TAF1(2), indicating that these residues could be considered as hot spots for designing effectively selective inhibitors toward BRD9 and TAF1(2). We also expect that this work could aid in obtaining a deeper understanding of the selectivity mechanism of inhibitors and provide theoretical guidance for the design of novel selective inhibitors targeting BRD9 and TAF1(2).

## Figures and Tables

**Figure 1 molecules-28-02583-f001:**
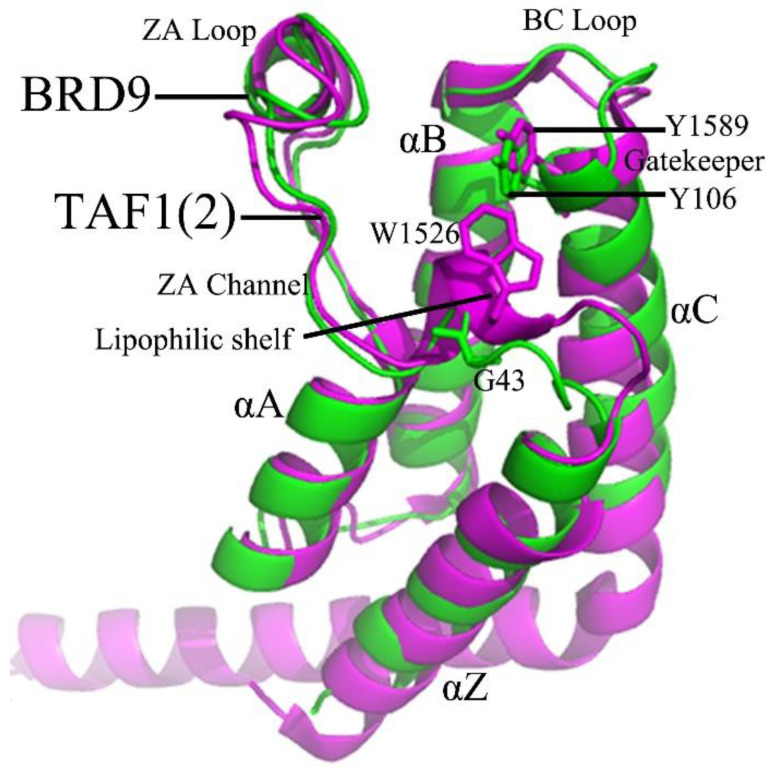
Tertiary structure superimposition of the BRD9 (PDB code 5I7X) and TAF1(2) (PDB code 5I29) bromodomains, in which BRD9 is displayed in green and TAF1(2) in magenta with cartoon modes. Notable residues include the gatekeeper (Y106 in BRD9 and Y1589 in TAF1(2)), the lipophilic shelf adjacent to the ZA channel (G43, F44, and F45 in BRD9), as well as W1526, P1527, and F1528 in TAF1(2).

**Figure 2 molecules-28-02583-f002:**
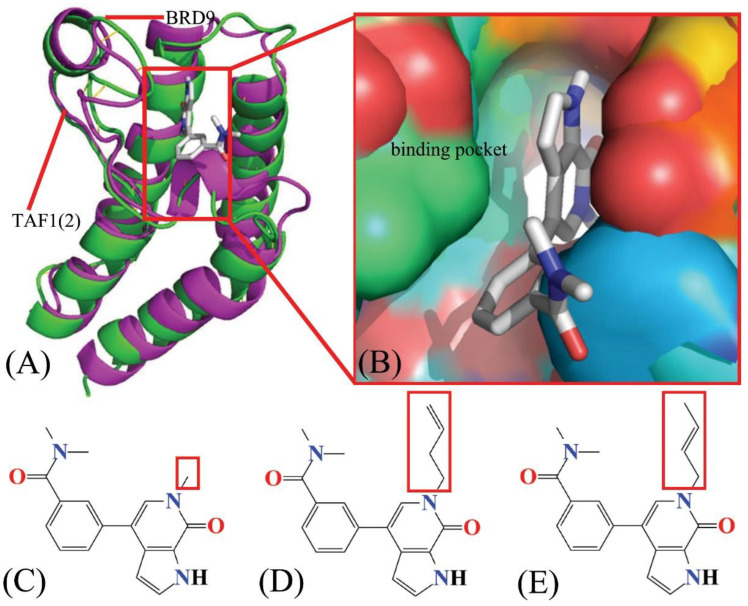
Molecular structures: (**A**) The superimposed structures of the 67B complex with BRD9 and TAF1(2), and 67B in complex with BRD9 (PDB code 5I7X) and 67B bound to TAF1(2) (PDB code 5I29), in which BRD9 is displayed in green and TAF1(2) in magenta with cartoon modes, (**B**) Binding pockets of BRD9 and TAF1(2) are shown in the surface modes and inhibitor in the stick mode, (**C**–**E**) correspond to the structures of 67B, 67C, and 69G, respectively, from which inhibitors are depicted in line forms.

**Figure 3 molecules-28-02583-f003:**
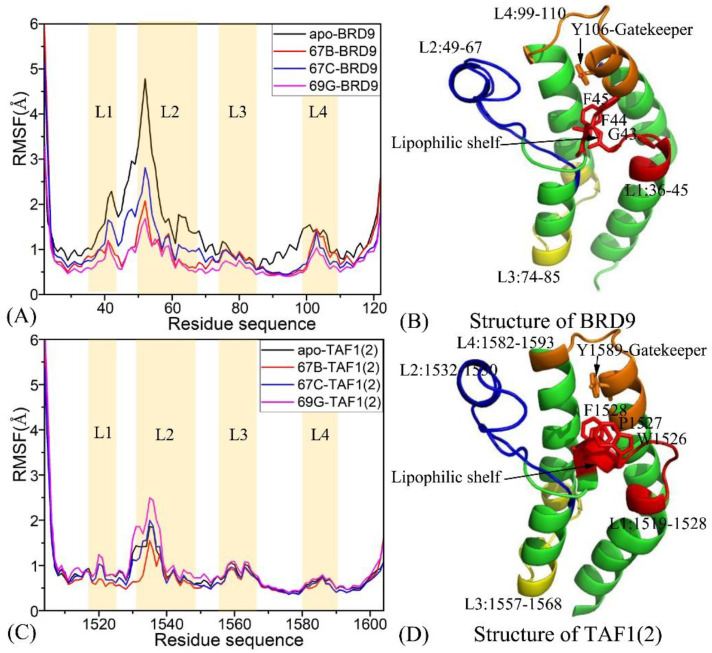
Root-mean-square fluctuations (RMSFs) of the C_α_ atoms in two proteins BRD9 and TAF1(2): (**A**) for BRD9 complexed with three inhibitors 67B, 67C, and 69G, (**B**) the structure of BRD9, (**C**) for TAF1(2) complexed with three inhibitors 67B, 67C and 69G, and (**D**) the structure of TAF1(2). The L1, L2, L3 and L4 are used to the regions with obvious changes of RMSFs.

**Figure 4 molecules-28-02583-f004:**
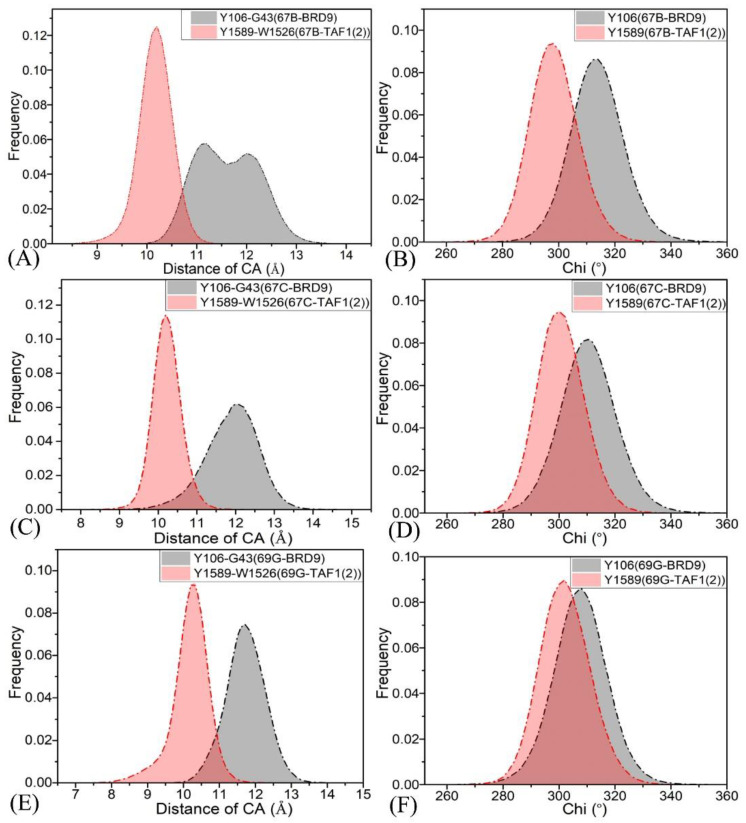
The frequency distribution of the Y106–G43 and Y1589–W1526 distances, and frequency distribution of the Chi dihedral angle of the side chain of Y106 in BRD9 and Y1589 in TAF1(2): (**A**) the frequency of distances in the 67B–BRD9 or TAF1(2) complexes, (**B**) the frequency distribution of Chi in the 67B–BRD9 or TAF1(2) complexes, (**C**) the frequency of distances in the 67C–BRD9 or TAF1(2) complexes, (**D**) the frequency distribution of Chi in the 67C–BRD9 or TAF1(2) complexes, (**E**) the frequency of distances in the 69G–BRD9 or TAF1(2) complexes, and (**F**) the frequency distribution of Chi in the 69G–BRD9 or TAF1(2) complexes.

**Figure 5 molecules-28-02583-f005:**
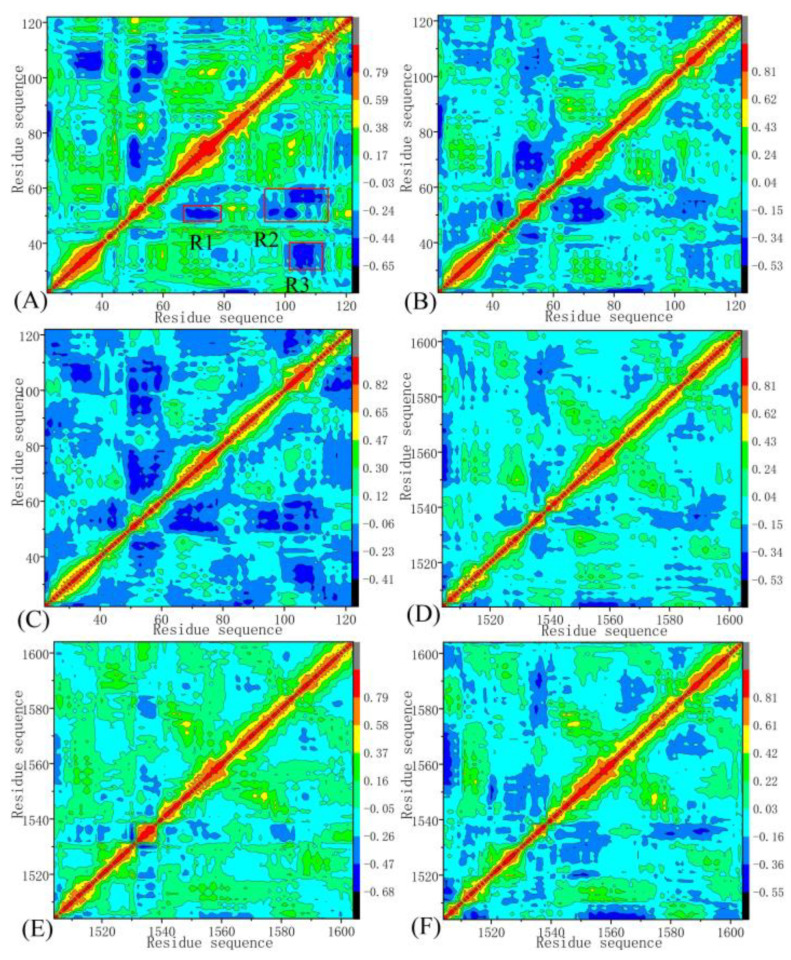
Dynamic cross-correlation maps computed by utilizing the coordinates of the $C_{\alpha }$ atoms around their mean positions recorded at the single joined trajectory: (**A**,**C**,**E**) BRD9 complexed with 67B, 67C, and 69G, respectively; (**B**,**D**,**F**) TAF1(2) complexed with 67B, 67C, and 69G, respectively.

**Figure 6 molecules-28-02583-f006:**
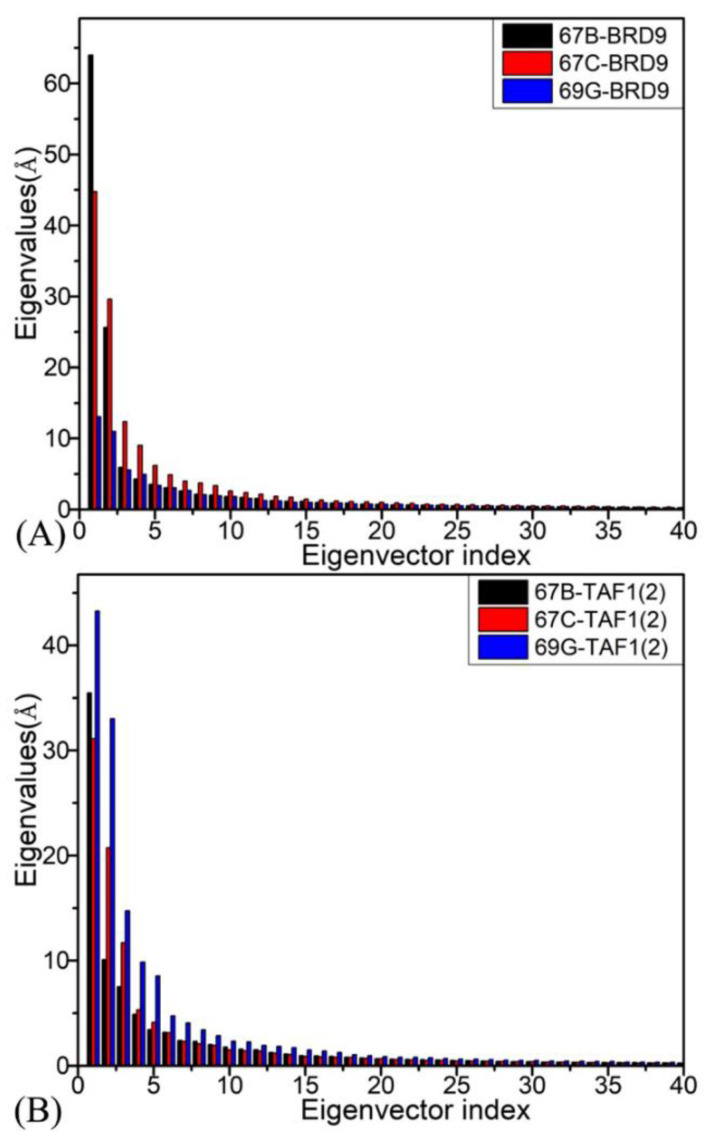
The function of the eigenvalues versus the eigenvector index extracted from PCA based on the single joined MSMD trajectory: (**A**) BRD9 and (**B**) TAF1(2) bonded with three inhibitors 67B, 67C, and 69G, respectively.

**Figure 7 molecules-28-02583-f007:**
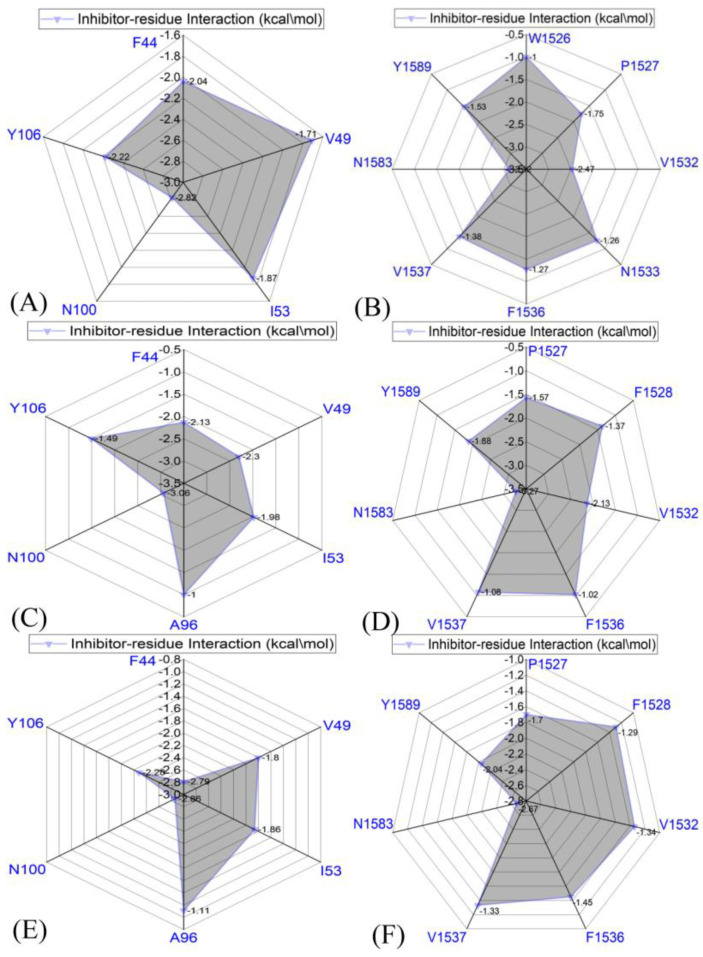
Inhibitor–residue interactions computed using the residue-based free energy decomposition method; only residues stronger than 1.0 kcal/mol are listed: (**A**) the 67B–BRD9 complex, (**B**) the 67B–TAF1(2) complex, (**C**) the 67C–BRD9 complex, (**D**) the 67C–TAF1(2) complex, (**E**) the 69G–BRD9 complex, and (**F**) the 69G–TAF1(2) complex.

**Figure 8 molecules-28-02583-f008:**
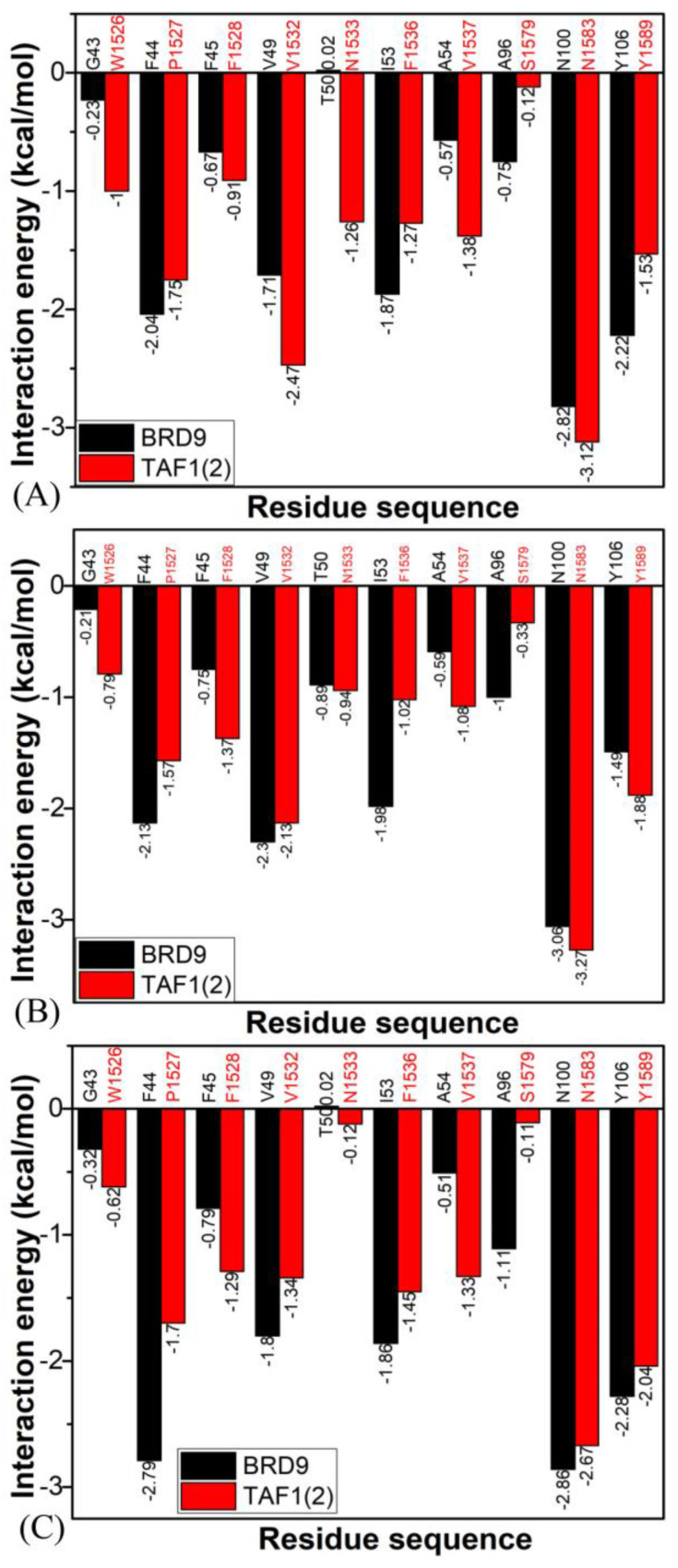
Interactions of inhibitors with important residues in BRD9 and TAF1(2): (**A**) 67B, (**B**) 67C, and (**C**) 69G.

**Figure 9 molecules-28-02583-f009:**
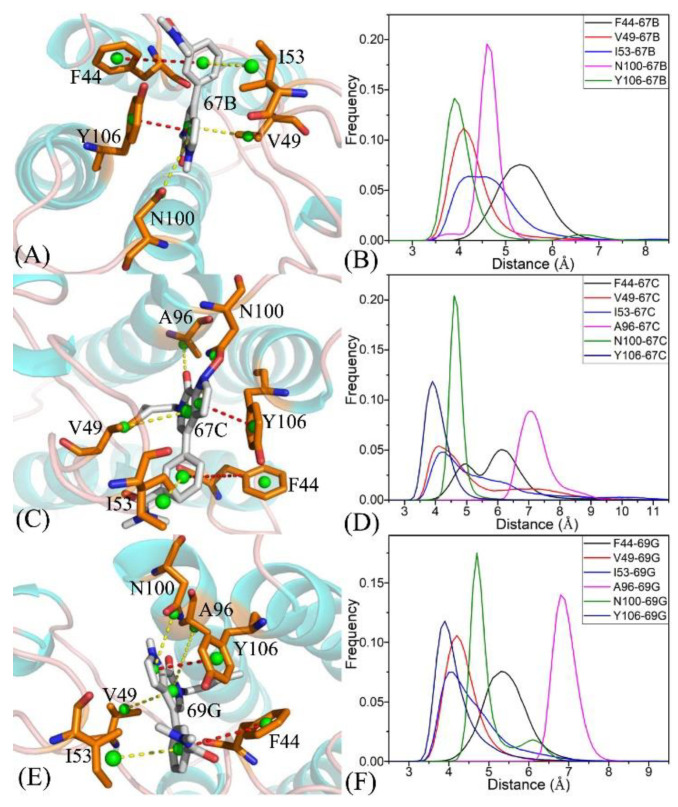
Hydrophobic interactions and frequency distributions of the distance involved in interactions of inhibitors with important residues: (**A**) the 67B–BRD9 complex, (**B**) RDF of 67B–BRD9, (**C**) the 67C–BRD9 complex, (**D**) RDF of 67C–BRD9, (**E**) the 69G–BRD9 complex, and (**F**) RDF of 69G–BRD9. The frequency of distances between atoms involving significant interactions were calculated by using the integrated MSMD trajectories of the last 900 ns. The yellow dashed lines describe the CH–π interactions and the red dashed lines indicate the π–π interactions.

**Figure 10 molecules-28-02583-f010:**
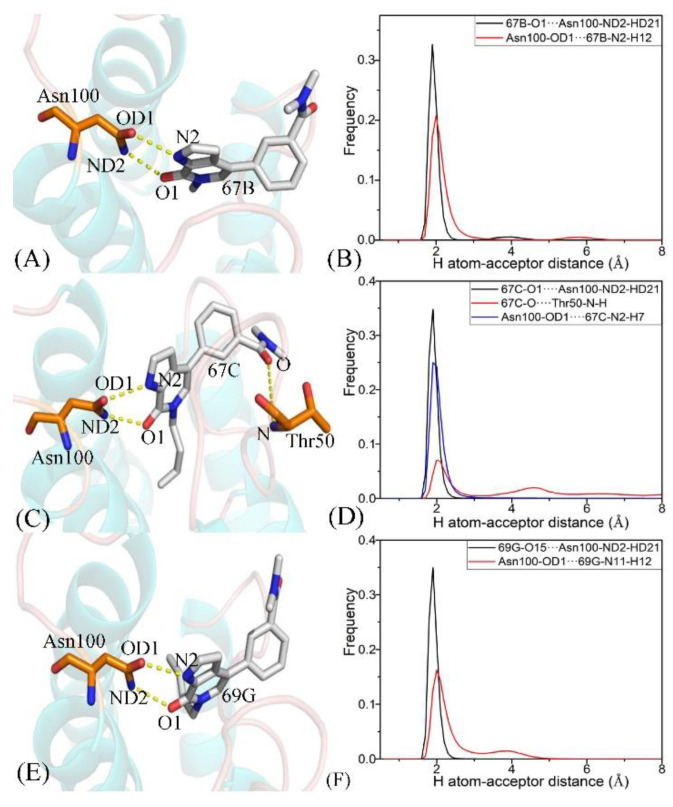
Hydrogen bonds and the corresponding radial distribution function (RDF) of the H–O distance between three inhibitors and key residues of BRD9: (**A**) the 67B–BRD9 complex, (**B**) RDF of H–O distances between 67B–O1 and Asn100-ND2-HD21, and 67B-N2-H12 and Asn100-OD1, (**C**) the 67C–BRD9 complex, (**D**) RDF of the H–O distances between 67C–O1 and Asn211-ND2-HD21, 67C-O, and Thr50-N-H, and Asn100-OD1 and 67C-N2-H7, (**E**) the 69G–BRD9 complex, and (**F**) RDF of the H–O distances between 69G–O15 and Asn100-ND2-HD21, and Asn100-OD1 and 69G-N11-H12.

**Figure 11 molecules-28-02583-f011:**
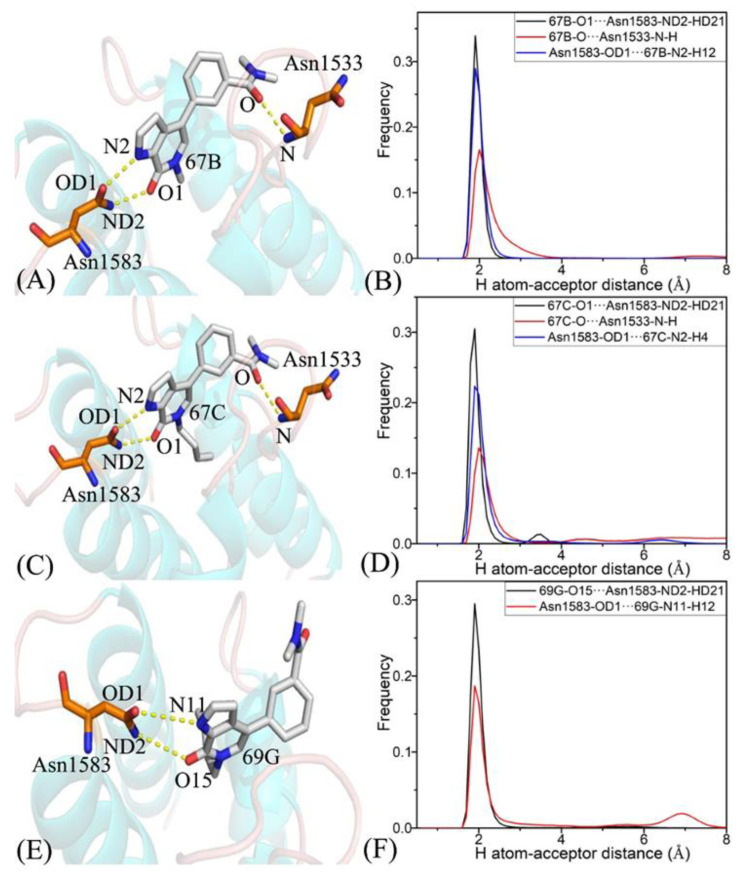
Hydrogen bonds and the corresponding radial distribution function (RDF) of the H–O distance between three inhibitors and key residues of TAF1(2): (**A**) the 67B–TAF1(2) complex, (**B**) RDF of H–O distances between 67B–O1 and Asn1583-ND2-HD21, 67B–O, and Asn1533-N-H, and Asn1583-OD1 and 67B-N2-H12, (**C**) the 67C–TAF1(2) complex, (**D**) RDF of the H–O distances between 67C–O1 and Asn1583-ND2-HD21, 67C–O, and Asn1533-N-H, and Asn1583-OD1 and 67C-N2-H4, (**E**) the 69G–TAF1(2) complex, and (**F**) RDF of the H–O distances between 69G–O15 and Asn1583-ND2-HD21, and Asn1583-OD1 and 69G-N11-H12.

**Figure 12 molecules-28-02583-f012:**
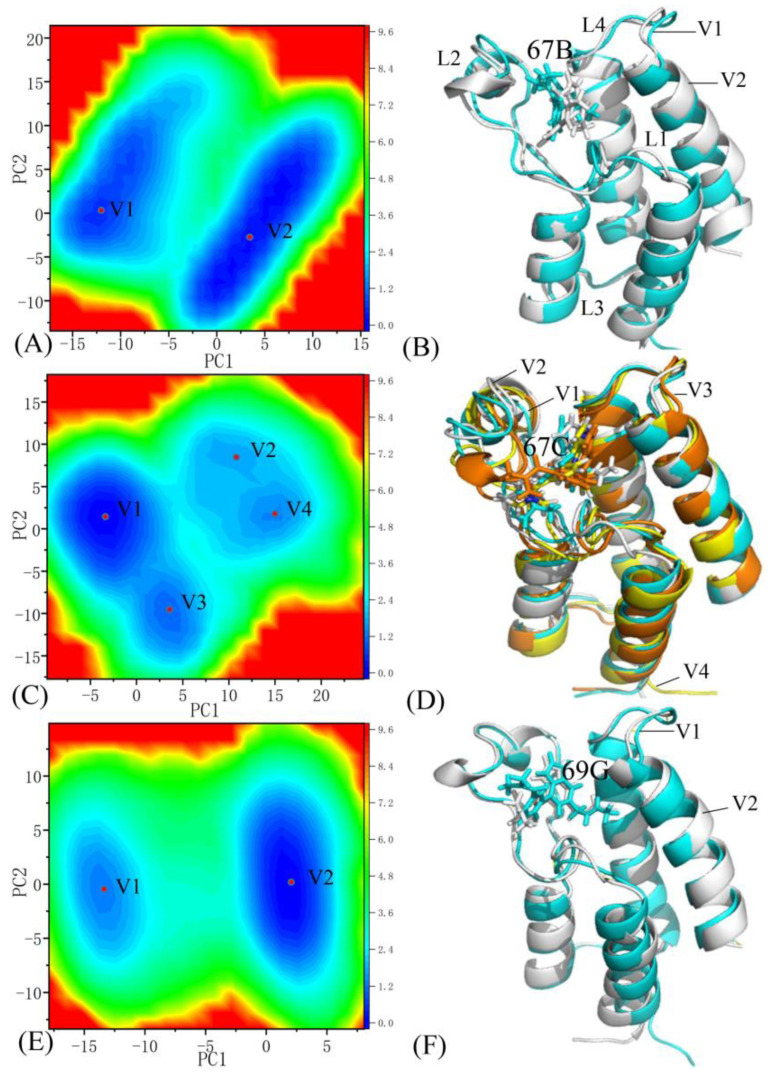
Free energy landscapes and structural information: (**A**,**C**,**E**) free energy landscapes of the 67B, 67C, and 69G-bound BRD9, respectively; (**B**,**D**,**F**) structural superimpositions of 67B, 67C, and 69G complexed with BRD9 situated at various energetic valleys, respectively. BRD9 and three inhibitors, 67B, 67C, and 69G, are depicted in cartoon and stick forms, respectively. The symbols V1, V2, V3, and V4 are labeled as the bottom of energy valleys. The L1, L2, L3 and L4 are used to the regions with obvious changes of structural flexibility.

**Figure 13 molecules-28-02583-f013:**
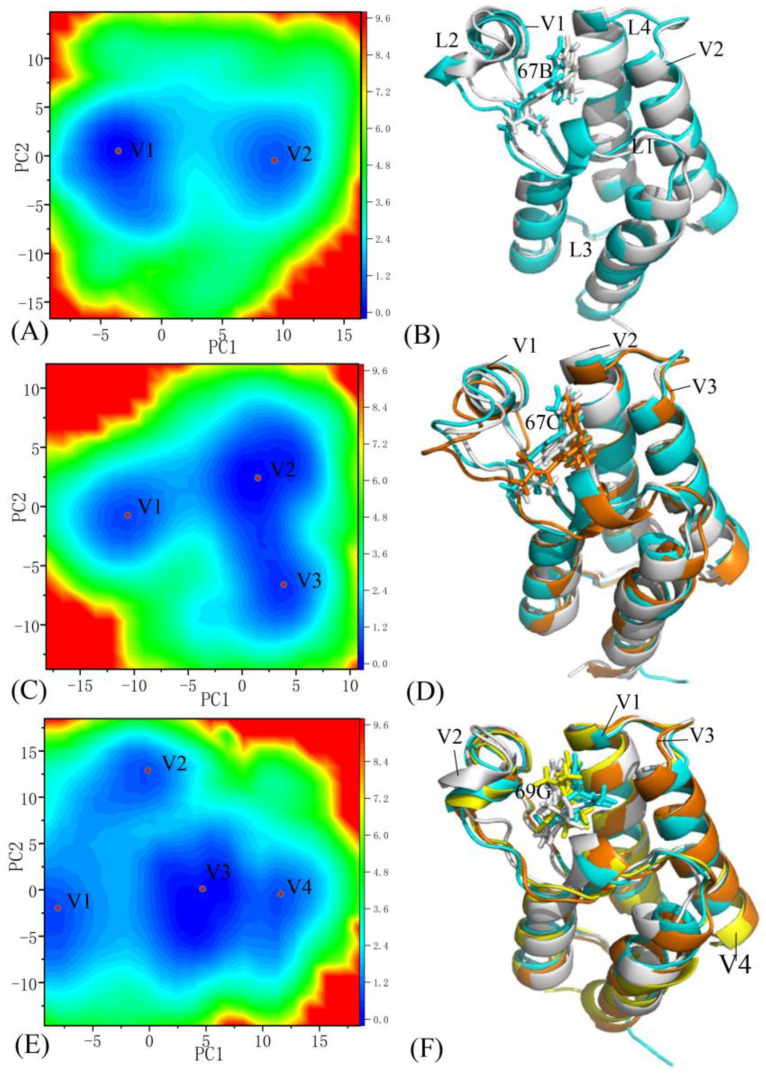
Free energy landscapes and structural information: (**A**,**C**,**E**) free energy landscapes of the 67B, 67C, and 69G-bound TAF1(2), respectively; (**B**,**D**,**F**) structural superimpositions of the 67B, 67C, and 69G complexed with TAF1(2), respectively, situated at different energetic valleys. TAF1(2) and three inhibitors, 67B, 67C, and 69G, are displayed in cartoon and stick forms, respectively.

**Table 1 molecules-28-02583-t001:** Binding affinities of small-molecule inhibitors to BRD9 and TAF1(2) computed with the MM-GBSA approach.

Components ^a^	67B−BRD9		67B−TAF1(2)		67C−BRD9		67C−TAF1(2)		69G−BRD9		69G−TAF1(2)	
	Mean	^b^ Sem	Mean	^b^ Sem	Mean	^b^ Sem	Mean	^b^ Sem	Mean	^b^ Sem	Mean	^b^ Sem
$\Delta E_{ele}$	−27.41	0.42	−34.82	0.36	−20.00	0.39	−31.69	0.45	−14.95	0.18	−16.73	0.43
$\Delta E_{vdW}$	−33.53	0.27	−35.93	0.25	−35.95	0.40	−38.99	0.31	−38.83	0.24	−36.04	0.24
$\Delta G_{gb}$	36.73	0.38	43.24	0.36	29.66	0.37	41.68	0.43	26.45	0.12	27.63	0.38
$\Delta G_{nonpol}$	−2.92	0.02	−3.29	0.02	−3.33	0.04	−3.67	0.02	−3.44	0.01	−3.29	0.02
^c^ $\Delta G_{ele+gb}$	9.32	0.40	8.42	0.36	9.66	0.38	9.99	0.44	11.5	0.15	10.9	0.41
^d^ $\Delta G_{vdW+nonpol}$	−36.45	0.14	−39.22	0.13	−39.28	0.22	−42.66	0.17	−42.27	0.13	−39.33	0.13
^e^ ΔH	−27.13	0.23	−30.80	0.23	−29.62	0.35	−32.67	0.31	−30.77	0.19	−28.43	0.21
−TΔS	16.08	0.77	17.02	0.64	20.59	0.69	18.65	0.60	18.19	0.73	17.62	0.71
$\Delta G_{bind}$	−11.05		−13.78		−9.03		−14.02		−12.58		−10.81	
IC_50_ (nM)	230		59		1400		46		160		410	
^f^ $\Delta G_{exp}$	−9.08		−9.89		−8.01		−10.0		−9.29		−8.73	

^a^ All components of the binding free energies are given in kcal/mol. ^b^ Standard errors of means (Sem). ^c^
$\Delta G_{ele+gb}=\Delta E_{ele}+\Delta G_{gb}$. ^d^ 
$\Delta G_{vdW+nonpol}=\Delta E_{vdW}+\Delta G_{nonpol}$. ^e^ ΔH = $\Delta E_{ele+gb}+\Delta E_{vdW+nonpol}$. ^f^ The experimental values were generated from the experimental IC_50_ values in [22] using the equation $\Delta G=-RTln{\mathrm{IC}}_{50}$.

**Table 2 molecules-28-02583-t002:** Hydrogen bonding interactions between inhibitors and BRD9 and TAF1(2) calculated using the CPPTRAJ program.

Complexes	Hydrogen Bonds	Distance/(Å) ^a^	Angle/(°) ^a^	Occupancy/(%) ^b^
67B–BRD9	67B-O1···Asn100-ND2-HD21 ^c^	2.85	159.98	94.64
	Asn100-OD1···67B-N2-H12	2.95	151.93	91.54
67B–TAF1(2)	67B-O1···Asn1583-ND2-HD21	2.87	163.93	99.72
	67B-O···Asn1533-N-H	3.05	160.73	82.41
	Asn1583-OD1···67B-N2-H12	2.91	163.09	99.42
67C–BRD9	67C-O1···Asn100-ND2-HD21	2.85	162.56	98.72
	67C-O···Thr50-N-H	3.00	150.92	29.81
	Asn100-OD1···67C-N2-H7	2.93	157.22	94.89
67C–TAF1(2)	67C-O1···Asn1583-ND2-HD21	2.85	164.30	93.97
	67C-O···Asn1533-N-H	3.04	162.59	63.61
	Asn1583-OD1···67C-N2-H4	2.95	163.10	88.87
69G–BRD9	69G-O15···Asn100-ND2-HD21	2.84	160.96	99.94
	Asn100-OD1···69G-N11-H12	3.01	155.82	78.93
69G–TAF1(2)	69G-O15···Asn1583-ND2-HD21	2.89	161.39	96.94
	Asn1583-OD1···69G-N11-H12	2.92	159.63	71.13

^a^ Hydrogen bonds were determined by the acceptor–donor atom distance of <3.5 Å and acceptor–H-donor angle of >120°. ^b^ Occupancy (%) was defined as the percentage of simulation time for which a specific hydrogen bond existed. ^c^ The full lines represent chemical bonds, and the dotted lines indicate hydrogen bonding interactions.

## Data Availability

Not applicable.

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
