# Peer review of "Deciphering Selectivity Mechanism of BRD9 and TAF1(2) toward Inhibitors Based on Multiple Short Molecular Dynamics Simulations and MM-GBSA Calculations"

_molecules, 2023, doi:10.3390/molecules28062583_

Round 1

Reviewer 1 Report

In their article, Wang and collaborators propose an interesting in silico study using molecular dynamics (MD) simulations to unravel molecular mechanisms associated to the binding of three known-compounds to two target interests: BRD9 and TAF1(2), two Bromodomains proteins.

I think that this paper deserves eventually to be publish in Molecules, but first, clarifications and modifications must be made to the article in its current form.

Major points:

1) The main criticism I have is that the authors tried to understand the impact of ligand binding to the two proteins, but no study about the apo form of the proteins was realized. By consequence, it is quite difficult to apprehend the difference between the structural and dynamical behavior of the proteins in apo and holo forms. Are the conformations sampled here exists also for the proteins alone or not? Is one of the ligand will stabilize one specific conformations in regard to the other ones?

In my opinion, authors must include some results regarding MD simulations on the apo-proteins. The ideal will be to realize also three replicates of 400 ns, but I recognize that all those calculations could take long time, so maybe one replicate could be enough in order to get some clue about the basal behavior of BRD9 and TAF1(2). Results could be inserted in supplementary materials.

2) The other major point concerns the behavior of the ligands itself. No clues about their motions are given until the last figures in which we see that one ligand could have four distinct conformations (67-C for BRD9 and G7-G for TAF1(2)). However, this is of great importance for the MM-GBSA analysis, in order to see if the four confirmations have different binding energy.  Table 1, presented as it is, suggest that the conformations of the ligands do not change during all the MDs, but it's not the case for two of them. Please, add some clues on the binding energy variations (if any) for the four conformations on the two concerned MDs and details about those conformational change. For example, does those four conformations are sampled during the same replicate or not? Is there a replicate in which there is only one conformational change?

Minor points:

1) Sequence of the two proteins are far more longer than the systems presented here. Could the authors justify their choice about only those selected portions for the study? I understand that the size could change a lot, but we cannot exclude that domain interactions could also have an importance and could change their results. Some comments must be add in the text.

2) In the introduction, details about the choice of the ligands and their activity must be insert. The fact that those ligands are already described and have known inhibitory activity for the two proteins comes very late in the main text.

3) There is a long discussion about DCCM and PCA analysis, but the figures associated are in the supplementary material. If this discussion is important, so those figures must be inserted in the main text and not in the supplementary materials, which are optional for the reader. Authors must make a choice here: either insert those figures in the main or significantly reduce the discussion about those figures in the main text and putting those details in the supplementary material.

4) On the contrary, table 2 is quite huge and the discussion about it is quite short. In my opinion, this table could be placed in the supplementary material.

5) How are chosen the representative structures of the several energetic valleys in figure 9 and 10? And how the free energy was calculated between the several states?

Author Response

Reviewer #1

In their article, Wang and collaborators propose an interesting in silico study using molecular dynamics (MD) simulations to unravel molecular mechanisms associated to the binding of three known-compounds to two target interests: BRD9 and TAF1(2), two Bromodomains proteins.

I think that this paper deserves eventually to be publish in Molecules, but first, clarifications and modifications must be made to the article in its current form.

Major points:

1) The main criticism I have is that the authors tried to understand the impact of ligand binding to the two proteins, but no study about the apo form of the proteins was realized. By consequence, it is quite difficult to apprehend the difference between the structural and dynamical behavior of the proteins in apo and holo forms. Are the conformations sampled here exists also for the proteins alone or not? Is one of the ligand will stabilize one specific conformations in regard to the other ones?

In my opinion, authors must include some results regarding MD simulations on the apo-proteins. The ideal will be to realize also three replicates of 400 ns, but I recognize that all those calculations could take long time, so maybe one replicate could be enough in order to get some clue about the basal behavior of BRD9 and TAF1(2). Results could be inserted in supplementary materials.

Reply:

Thank you very much for this valuable suggestion.

According to your suggestion, we have implemented three independent simulations of 400 ns on the apo BRD9 and TAF1(2), respectively. Moreover, we calculated the RMSDs and RMSFs of two proteins BRD9 and TAF1(2), shown in Figure S1 and Figure 3A and 3C. Meanwhile we discussed the results and revised our manuscript. The revised contents are highlighted in the red.

“On the whole, the structural fluctuation range of the apo BRD9 is higher than that of the inhibitor-BED9 complexes while the structural fluctuation extent of the apo TAF1(2) is similar to that inhibitor-TAF1(2) complexes (Figure S1A and S1B).”

“For the BRD9-related systems, binding of three inhibitors weaken the struc-tural flexibility of BRD9, especially for the ZA-loop (Figure 3A and 3B). However, for the TAF1(2)-related systems, binding of 67B and 67C slightly reduces the structural flexibility of TAF1(2) while the presence of 69G strengthens that of TAF1(2), in particular for the ZA-loop (Figure 3C and 3D).”

       Thank you again for this valuable suggestion.

2) The other major point concerns the behavior of the ligands itself. No clues about their motions are given until the last figures in which we see that one ligand could have four distinct conformations (67-C for BRD9 and 69-G for TAF1(2)). However, this is of great importance for the MM-GBSA analysis, in order to see if the four confirmations have different binding energy.  Table 1, presented as it is, suggest that the conformations of the ligands do not change during all the MDs, but it's not the case for two of them. Please, add some clues on the binding energy variations (if any) for the four conformations on the two concerned MDs and details about those conformational change. For example, does those four conformations are sampled during the same replicate or not? Is there a replicate in which there is only one conformational change?

Reply:

       Thank you very much for this valuable suggestion. Free energy landscapes reveal that three inhibitors have different binding poses in the binding pocket of two proteins BRD9 and TAF1(2), but the structural superimpositions of inhibitors located at different energy valleys suggests that inhibitors only minor deviations, which possibly generates minor effect on binding of inhibitors to BRD9 and TAF1(2). In order to check this issue, we calculated RMSDs of non-hydrogen atoms from three inhibitors and the corresponding information is added at the supporting information Figure S2. The results show that the structures of inhibitors do not generate obvious fluctuations, which implies that three inhibitors are well kept at binding pocket. Thus the calculations of MM-GBSA are reliable.

Thank you again for this valuable comment and suggestion.

Minor points:

  • Sequence of the two proteins are far more longer than the systems presented here. Could the authors justify their choice about only those selected portions for the study? I understand that the size could change a lot, but we cannot exclude that domain interactions could also have an importance and could change their results. Some comments must be add in the text.

Reply:

Firstly, we agree with you, the sequence of the two proteins are different from each other. In fact, the sequence of 67B in complex with BRD9 (PDB code 5I7X) is 22-122, while the sequence of 67B bound to TAF1(2) (PDB code 5I29) is 1500-1638. These two sequence are different, which is difficult to investigate the ligand affinity to these two protein. In this work, we choose two identity sequence to decipher selectivity mechanism of BRD9 and TAF1(2) toward three inhibitors. The structural superimposition shown in Figure 1 reveal sequence difference, thus we selected the same residue length to perform the current study.

     According to your valuable suggestion, we revised our manuscript and the revised parts are highlighted with the red.

     Thank you again for this valuable suggestion and comment.

“Due to the difference in the number of residues from BRD9 and TAF1(2), the residues 22-122 in BRD9 and the residues 1504-1604 in TAF1(2) were used as the starting models of MD simulations.”

  • In the introduction, details about the choice of the ligands and their activity must be insert. The fact that those ligands are already described and have known inhibitory activity for the two proteins comes very late in the main text.

Reply:

       Thank you very much for your valuable comment.

       By following this comment, we have revised our manuscript and the revised parts are highlighted in the red.

“Based on important target roles of BRD9 and TAF1(2) in drug design toward human cancers, Crawford and the coworkers solved the crystal structures of the 67B- and 69G-bound BRD9 as well as the 67B- and 67C-associated TAF1(2) [22]. Despite high similar structures shared by 67B, 67C and 69G (Figure 2C-2E), three inhibitors have different binding affinity to BRD9 and TAF1(2), with IC50 values of 230/59 nM, 1400/46 nM, and 160/410 nM for BRD9/TAF1(2), respectively. Therefore, it is essential to clarify binding selectivity of inhibitors to BRD9 and TAF1(2) and conformational changes of two proteins caused by inhibitor bindings for drug design for anti-cancer treatment.”

  • There is a long discussion about DCCM and PCA analysis, but the figures associated are in the supplementary material. If this discussion is important, so those figures must be inserted in the main text and not in the supplementary materials, which are optional for the reader. Authors must make a choice here: either insert those figures in the main or significantly reduce the discussion about those figures in the main text and putting those details in the supplementary material.

Reply:

       Thank you very much for your valuable suggestion, we have moved two figures on DCCM and PCA to the main text.

  • On the contrary, table 2 is quite huge and the discussion about it is quite short. In my opinion, this table could be placed in the supplementary material.

Reply:

Thank you very much for this valuable suggestion.

According to your valuable suggestion, we have moved the table 2 to the supplementary material and revised it as Table S1.

  • How are chosen the representative structures of the several energetic valleys in figure 9 and 10? And how the free energy was calculated between the several states?

Reply:

       Thank you very much for your positive comments and the efforts paid by you in reviewing our manuscript.

Firstly, we used the principal components PC1 and PC2 as reaction coordinates to calculate free energy and the calculated free energies are scaled in color bar. Through those free energy landscapes, we identified several low energy states (Valleys), then we determined the coordinates of the valley bottom. Finally, we used these coordinates of the valley bottom to extract the representative structures from the MD trajectories.

Finally, thank you very much for your valuable suggestion and effectors paid by you in reviewing our manuscript.

Reviewer 2 Report

On reading the manuscript, I see that the study is well done and should deserve publication. Unfortunately, it cannot be accepted for publication in the present form. The organization of the manuscript renders difficult the understanding for interested but not so expert readers.

Important information, for instance, the existence of the crystal structures of the two bromodomains complexed with a couple of the inhibitors, is provided in the Materials and Methods section; this has to be provided in the Introduction because this is the reason why the MD simulations can be carried out in a reliable manner. General structural features of the bromodomains are not provided at all (i.e., their four-helix bundle fold), and the structural differences between the two selected bromodomains are not explained; a Figure comparing the two bromodomains without inhibitors could be very useful and should be added in the Introduction. Note, in this respect, that Figures 6-10 are well done and are very clear.

The other weak point is related to the English. The latter is not bad in general, but several sentences are written in a poor English or with errors. This can create misunderstandings.

The authors have to perform a major revision of the manuscript following the above suggestions.

Author Response

Reviewer #2:

On reading the manuscript, I see that the study is well done and should deserve publication. Unfortunately, it cannot be accepted for publication in the present form. The organization of the manuscript renders difficult the understanding for interested but not so expert readers.

Important information, for instance, the existence of the crystal structures of the two bromodomains complexed with a couple of the inhibitors, is provided in the Materials and Methods section; this has to be provided in the Introduction because this is the reason why the MD simulations can be carried out in a reliable manner. General structural features of the bromodomains are not provided at all (i.e., their four-helix bundle fold), and the structural differences between the two selected bromodomains are not explained; a Figure comparing the two bromodomains without inhibitors could be very useful and should be added in the Introduction. Note, in this respect, that Figures 6-10 are well done and are very clear.

Reply:

       Thank you very much for this valuable suggestion.

According to your valuable suggestion, we added the Figure 1 in the Introduction, which compared the tertiary structure of the BRD9 and TAF1(2) and the revised contents are highlighted in the red.

“Figure 1. Tertiary structure of the BRD9 (PDB code 5I7X) and TAF1(2) (PDB code 5I29) bromodomains, in which BRD9 is displayed in green and TAF1(2) in magenta with cartoon modes. Notable residues include the gatekeeper (Y106 in BRD9 and Y1589 in TAF1(2)), the lipophilic shelf adjacent to the ZA channel (G43, F44, F45 in BRD9), as well as (W1526, P1527, and F1528 in TAF1(2)).”

"Based on important target roles of BRD9 and TAF1(2) in drug design toward human cancers, Crawford and the coworkers solved the crystal structures of the 67B- and 69G-bound BRD9 as well as the 67B- and 67C-associated TAF1(2) [22]. Despite high similar structures shared by 67B, 67C and 69G (Figure 2C-2E), three inhibitors have different binding affinity to BRD9 and TAF1(2), with IC50 values of 230/59 nM, 1400/46 nM, and 160/410 nM for BRD9/TAF1(2), respectively. Therefore, it is essential to clarify binding selectivity of inhibitors to BRD9 and TAF1(2) and conformational changes of two proteins caused by inhibitor bindings for drug design for anti-cancer treatment.”

The other weak point is related to the English. The latter is not bad in general, but several sentences are written in a poor English or with errors. This can create misunderstandings. The authors have to perform a major revision of the manuscript following the above suggestions.

Reply:

Thank you very much for this kindly reminding and suggestion. We have revised our manuscript and the English description with help of an expert whose mother language is English.

Finally, thank you very much for your valuable suggestion and the efforts in reviewing our manuscript.

Reviewer 3 Report

The manuscript molecules-2236310 reports MD simulations aimed at studying the binding of three ligands (67B, 67C and 69G) to two drug targets (BRD9 and TAF1(2)), to understand the molecular basis of their anchoring. In the following, I will provide suggestions on (1) how to improve a bit the simulation methods, (2) deleting or rearranging a few parts of the text that are not appropriate, and (3) a list of minor revisions.

1.

- (line 581), “results imply that the MM-GBSA approach can provide more reasonable conclusions [94,95]”. Most of the time the two variants give similar results, and the point is to find a trade-off between accuracy and efficiency. See https://doi.org/10.1016/B978-0-12-819132-3.00014-2 for a more in-depth discussion and as a reference.

- (line 588) “third element is solved based on the GB model developed by Onufriev et al.”. Please specify which scheme (probably GB^OBC1), because two models are proposed in reference 96 (see equations 7 and 8), and this is an important point.

- (line 534) “TIP3P model are utilized to solve” -> “to solvate”. (line 559) “heavy atoms were confined with the SHAKE numerical integration algorithm” -> “were constrained”. (line 562) “reasonable cut-off value” -> delete “reasonable” because other similar values may be equally reasonable.

2.

- (line 52) “The BRD9 and TAF1(2), structurally sharing four left-handed α-helixes (αA, αB, αC, αZ) constituting up- and-down four-helix bundles, which form two loops between helixes αA and αZ (ZA loop) and αB and αC (BC loop), respectively [21]”. The paragraph starting here has a different indentation and tone compared to the rest, in between a caption of Figure 1 and an ordinary paragraph of text (please note that the sentence has no verb, resembling in this a caption). I believe this part should be adjusted in the form of a common paragraph. Also note that Figure 1 is not cited in the text until much later, so it should be cited here. As a side note, the plural of “helix” is “helices”.

- (line 337) “The 67B-bound BRD9 against the 67B-bound TAF1(2):”. This and other similar expressions are in several places, and in my opinion it would be much more elegant if they were transformed into sub-sub-sections (in this example, it would be a sub-sub-section of sub-section 2.4).

- (line 659) “Please turn to the CRediT taxonomy for the term explanation. Authorship must be limited to those who have contributed substantially to the work reported”. Please delete this part.

3.

- (line 17) “enthalpy alterations” -> “enthalpic contributions”

- (line 76) “etc.” -> please avoid.

- (line 117) “One can see that three inhibitors 67B, 67C and 69G are share an intensely similar structure except for the red rectangle. It is conducive to uncover”. Please change “intensely” and “uncover”, and avoid “the red rectangle” without an explicit reference to the figure.

- (line 164) “highly bigger”, (line 167) “incredibly rigid”, (line 168) “intensely flexible”. Do not use these expressions.

- (line 178 and 191) “the Chi (in degree) dihedral angle” -> “Chi dihedral angle (in degrees)”, and note that “(in degrees)” is better omitted.

- (line 193) “distance of BRD9-67B complex are distributed in 11.1 and 12.0 Å” -> “in the range between”.

- (line 197) “the hot interaction site volume”, (line 202) “hot interaction spot regions”: the correct expression is “hot spot”, either use it or do not use “hot” in other expressions.

- (line 210) “(dark blue and blue)” -> “and light blue” or “and plain blue”.

- (line 241) “also alter movement strengthen of these three loops”, the expression in unclear.

- (line 289) “kca/mol” -> “kcal/mol”.

- (line 292) “67B prefers associating with TAF1(2) to binding to BRD9”; do not use this expression, because it is not a competitive binding where both targets are simultaneously present.

Author Response

Reviewer #3:

The manuscript molecules-2236310 reports MD simulations aimed at studying the binding of three ligands (67B, 67C and 69G) to two drug targets (BRD9 and TAF1(2)), to understand the molecular basis of their anchoring. In the following, I will provide suggestions on (1) how to improve a bit the simulation methods, (2) deleting or rearranging a few parts of the text that are not appropriate, and (3) a list of minor revisions.

1.

- (line 581), “results imply that the MM-GBSA approach can provide more reasonable conclusions [94,95]”. Most of the time the two variants give similar results, and the point is to find a trade-off between accuracy and efficiency. See https://doi.org/10.1016/B978-0-12-819132-3.00014-2 for a more in-depth discussion and as a reference.

Reply:

Thank you very much for this valuable suggestion.

The references provided by the reviewer are very valuable, which better enriches our works. Thus, we mention this work in our manuscript.

“It is of high significance to achieve a good trade-off between the accuracy and efficiency in calculations of inhibitor-target binding free energies for drug development, which has been discussed by Rizzuti et al. in their work [90].”

  1. Rizzuti B.;Grande F. Virtual screening in drug discovery: a precious tool for a still-demanding challenge. Protein Homeostasis Diseases. Academic Press, 2020, 309–327.

- (line 588) “third element is solved based on the GB model developed by Onufriev et al.”. Please specify which scheme (probably GB^OBC1), because two models are proposed in reference 96 (see equations 7 and 8), and this is an important point.

Reply:

Thank you very much for this valuable suggestion, in our calculations, the GB-OBCI model is used. According to this suggestion, we revised our manuscript.

“The third element is solved based on the GB-OBCI model GB model developed by Onufriev et al.”

- (line 534) “TIP3P model are utilized to solve” -> “to solvate”. (line 559) “heavy atoms were confined with the SHAKE numerical integration algorithm” -> “were constrained”. (line 562) “reasonable cut-off value” -> delete “reasonable” because other similar values may be equally reasonable.

Reply:

 Thank you very much for this kindly reminding and suggestion.

According to your valuable suggestion, we corrected errors in our manuscript.

2.

- (line 52) “The BRD9 and TAF1(2), structurally sharing four left-handed α-helixes (αA, αB, αC, αZ) constituting up- and-down four-helix bundles, which form two loops between helixes αA and αZ (ZA loop) and αB and αC (BC loop), respectively [21]”. The paragraph starting here has a different indentation and tone compared to the rest, in between a caption of Figure 1 and an ordinary paragraph of text (please note that the sentence has no verb, resembling in this a caption). I believe this part should be adjusted in the form of a common paragraph. Also note that Figure 1 is not cited in the text until much later, so it should be cited here. As a side note, the plural of “helix” is “helices”.

Reply:

Thank you very much for this kindly reminding and valuable suggestion.

We added a new figure in the introduction to depict tertiary structure of the BRD9 and TAF1(2) bromodomains and changed the errors in our revised manuscript.

- (line 337) “The 67B-bound BRD9 against the 67B-bound TAF1(2):”. This and other similar expressions are in several places, and in my opinion it would be much more elegant if they were transformed into sub-sub-sections (in this example, it would be a sub-sub-section of sub-section 2.4).

Reply:

 Thank you very much for this kindly reminding and suggestion.

According to your valuable suggestion, we have changed these descriptions to sub-sub-sections in our manuscript.

- (line 659) “Please turn to the CRediT taxonomy for the term explanation. Authorship must be limited to those who have contributed substantially to the work reported”. Please delete this part.

 Reply:

 Thank you very much for this kindly reminding and suggestion.

According to your valuable suggestion, we have deleted this part in our manuscript.

3.

- (line 17) “enthalpy alterations” -> “enthalpic contributions”

- (line 76) “etc.” -> please avoid.

- (line 117) “One can see that three inhibitors 67B, 67C and 69G are share an intensely similar structure except for the red rectangle. It is conducive to uncover”. Please change “intensely” and “uncover”, and avoid “the red rectangle” without an explicit reference to the figure.

- (line 164) “highly bigger”, (line 167) “incredibly rigid”, (line 168) “intensely flexible”. Do not use these expressions.

- (line 178 and 191) “the Chi (in degree) dihedral angle” -> “Chi dihedral angle (in degrees)”, and note that “(in degrees)” is better omitted.

- (line 193) “distance of BRD9-67B complex are distributed in 11.1 and 12.0 Å” -> “in the range between”.

- (line 197) “the hot interaction site volume”, (line 202) “hot interaction spot regions”: the correct expression is “hot spot”, either use it or do not use “hot” in other expressions.

- (line 210) “(dark blue and blue)” -> “and light blue” or “and plain blue”.

- (line 241) “also alter movement strengthen of these three loops”, the expression in unclear.

- (line 289) “kca/mol” -> “kcal/mol”.

- (line 292) “67B prefers associating with TAF1(2) to binding to BRD9”; do not use this expression, because it is not a competitive binding where both targets are simultaneously present.

Reply:

       Thank you very much for this valuable suggestion.

According to your suggestion, we have changes these errors in our manuscript.

In the end, thanks you very much again for your valuable suggestions and the efforts paid by you in reviewing our manuscript. We hope that our modifications could meet your requirements sincerely.

Sincerely yours

Lifei Wang

Round 2

Reviewer 1 Report

The authors have greatly improved their article. it is now suitable for publication

Reviewer 2 Report

I am satisfied with the performed revision.